# Implementation of a sample design for a survey of program participants using time-location sampling

**Yumiko Siegfried**©*©, **Jill M. DeMatteis**©, **Bibi Gollapudi**©

Westat, Rockville, Maryland, United States of America

© These authors contributed equally to this work.

* yumikosiegfried@westat.com

**Data Availability Statement:** This manuscript uses WIC ITFPS-2 sample management data. For more information on these data, please contact OPSDataRequests@usda.gov.

## Abstract

To assess the feeding practices and behaviors of women and young children participating in the Special Supplemental Nutrition Program for Women, Infants, and Children (WIC), USDA currently funds the longitudinal WIC Infant and Toddler Feeding Practices Study-2 (WIC ITFPS-2). In 2013, the study used time-location sampling (TLS) to enroll a cohort of infants who participated in WIC around birth. The children are subsequently followed across their first six years of life, regardless of their participation in WIC, with an additional follow-up at age nine years. A woman may enroll her child in WIC either during pregnancy or postpartum. For this study, a representative sample of infants enrolled in WIC was desired. Because the associations between WIC prenatal support and education and feeding practices and behaviors are substantively important to this study, the sample needed to include both women enrolling their children prenatally and women enrolling their children postnatally. For prenatal WIC enrollees, we attempted to complete a prenatal interview with the mother prior to the child's birth. This paper describes the TLS approach used and the challenges addressed in implementation of the sample design and selection for the WIC ITFPS-2. Our approach generated a probability sample (subject to site geographic and size exclusions) using a stratified, multistage design, but there were challenges at each stage of selection. First, a WIC site was selected, and then newly enrolled WIC participants were sampled within selected sites during predetermined recruitment windows based on the site's average flow of new WIC enrollees. We discuss issues faced, including overcoming incomplete lists of individual WIC sites and discrepancies between projected new WIC enrollment counts and actual flow of new WIC enrollments during the recruitment period.

## Introduction

In designing a sample for a survey of program participants, the ideal situation is to have a current census of program participants, with contact information, that serves as the sampling frame. When that is not possible, alternatives that may be considered include screening a general population sample to identify program participants; using a nonprobability sample such

**Funding:** Funding for the WIC ITFPS-2 was provided to Westat by the Food and Nutrition Service of the U.S. Department of Agriculture. Funding for the development of this manuscript was provided by Westat.

**Competing interests:** The authors have declared that no competing interests exist.

as network sampling [1] or respondent-driven sampling [2]; or using time-location sampling (TLS) [3–6].

The basic objective of TLS (also referred to as "venue-based sampling" and "time-space sampling") is to sample individuals at locations they visit. Ideally, to mitigate concerns about undercoverage bias, a large proportion of the target population would visit the specific type of location. The sample design for a time-location sample involves first sampling the location, then sampling a period of time, then sampling individuals who visit the location during the sampled period of time. In general, TLS is used to sample members of a rare population, and common applications of TLS include studies of individuals participating in illegal activities or stigmatized individuals, such as illicit drug users, men who have sex with men, and female sex workers.

There are several potential challenges with TLS. First, for TLS to be viable, a type of location that is visited by a substantial proportion of the target population must be determined. Once the type of location has been identified, a comprehensive list of such locations must be obtained. Second, for each sampled location, it is necessary to identify the hours of operation and estimate the flow rates (the average rate at which individuals visit the location). Third, for each location, once a time period has been sampled, procedures must be implemented (generally on-site at the location) to identify and sample eligible individuals. The on-site sampling approach generally involves counting eligible individuals and selecting a systematic sample. If study recruitment is to be done on-site, then a minimum of two data collectors (one to count and select the sample and another to recruit sampled individuals into the study) is needed.

Identifying hours of operation and estimating flow rates is often best done by contacting the locations themselves; however, that poses challenges in that locations may refuse to cooperate or might not have the information available, particularly the flow rates. When flow rates must be estimated, it may be necessary to provide tools to assist the location representatives with this. Inaccuracies in the estimated flow rates may result in understaffing of the recruitment effort (if the actual flow rate is higher than estimated) or insufficient sample yield (if the actual flow rate is lower than estimated).

Following the sampling of locations and time periods, the sampling of individuals may be done either by obtaining a list of visitors to the location during the specified time period (if available) and sampling from that list or by conducting sampling on-site. Either approach requires gaining the cooperation of the locations. With the on-site sampling approach, changes or variations in days and/or hours of operation and in visitor counts may wreak havoc on the study recruitment operations and introduce variation into the number sampled for the study. Additionally, for individuals who visit multiple locations or visit a given location multiple times, the multiplicity must be captured (via questions included in the survey) and adjusted for in computing survey weights designed to yield unbiased estimates.

In this paper, we describe an application of TLS to select a national sample of Federal nutrition assistance program participants (with restrictions based on geography and size of the site). Administered by the U.S. Department of Agriculture (USDA), the Special Supplemental Nutrition Program for Women, Infants, and Children (WIC) is the premier public health nutrition program for low-income pregnant and postpartum women and children ages birth to five years. Because all participants are nutritionally at-risk, WIC provides participants with three benefits: a monthly supplemental food package, nutrition education including breastfeeding support, and healthcare referrals. Our application involves the sampling of new enrollees into WIC; additional details regarding sampling and recruitment into this study may be found in Appendix B of WIC Infant and Toddler Feeding Practices Study-2: Fourth year report [7].

For probability samples, each individual must have a known, positive probability of selection. In many applications of TLS, a challenge is identifying for inclusion in the study types of

locations that are frequented by a large proportion of the target population (e.g., [8, 9]); we did not face that challenge, as at the time of recruitment for this study, all participants were required to enroll in WIC in-person by visiting a WIC site. A participant could either enroll herself in WIC prenatally (in which case, the enrollment is extended to the child(ren) when the birth occurs) or enroll her child(ren) postnatally. For TLS, computing an individual's probability of selection requires knowledge of the frequency at which they visit the locations, and accounting for these multiplicities when computing individuals' probabilities of selection; because our application covers new WIC enrollees, each participant is uniquely identified (when they first enroll the child in WIC), and there is no possibility of multiplicity. Thus, in contrast to many previous applications of TLS, which have produced non-probability samples or probability samples with low coverage of the target population, our approach was designed to yield a probability sample of new enrollees (subject to exclusions based on geographic and site size criteria).

## Methods

Sponsored by USDA Food and Nutrition Service (FNS), the second WIC Infant and Toddler Feeding Practices Study (WIC ITFPS-2) is a longitudinal cohort study that follows mothers and their young children from the time the mother or baby was enrolled in WIC, either prenatally or within 2.5 months after birth, through child age 6 with an additional follow-up at age 9. The study recruitment period started in July 2013 and ended in November 2013, and the baseline interview was a prenatal interview for prenatal WIC enrollees, and either a 1- or 3-month interview for postnatal WIC enrollees.

Human subjects' protections for the study are overseen by 17 Institutional Review Boards (IRBs), including: Westat; state Department of Health IRBs in CA, CT, FL, GA, LA, MD, MI, NY, OH, OK, PA, SC, TN, and TX; and local IRBs at Arrowhead Regional Medical Center in San Bernardino, CA, and Los Angeles Biomedical Research Institute at Harbor-UCLA Medical Center, CA.

Caregivers, typically pregnant women or new mothers, were eligible for the study if they were at least 16 years of age, spoke English or Spanish, were pregnant or had a child less than 2.5 months of age, and were enrolling in WIC for the first time for the pregnancy or child. Herein, we refer to caregivers enrolling in WIC for the first time for a specific pregnancy or child as "new enrollees" despite the fact that they may have enrolled in WIC for a previous pregnancy or child.

For participants recruited in person, the Westat Recruiter provided the consent form for the participant's review and signature, and a copy was provided to the participants for their records. For participants recruited over the phone, the consent form was read in its entirety and consent was obtained verbally. The consent form was mailed to those participants and they were asked to sign and return one copy. However, we received a waiver for signed consent from the overseeing IRBs, so participants who consented verbally continued in the study even if they did not return a signed consent form. For minor participants (the minimum age is 16), we obtained consent from the parents/guardians and assent from the participants. All participants consented verbally, and signed consent forms were obtained for 85 percent of study participants.

Following enrollment in the study, interviews were administered, regardless of whether the child continued to participate in WIC, beginning with the prenatal interview (for prenatal WIC enrollees), the 1- and/or 3-month interview, every two months through the 15-month interview, another interview at 18 months, then every six months through the 60-month interview, and finally again at 72 months of age (see S1 Fig). A nine-year-old follow-up is being

conducted in 2022–23. A total of 4,367 women/infants were enrolled in the study, and the response rate through the study enrollment was 82 percent. The response rates for specific cross-sectional interviews through age 72 months ranged from 58 percent to 88 percent, conditional on study enrollment. Also included in the study were periodic measurements of the child's length (or height) and weight. The considerations for analyses involving the measurement data are not discussed here, but are similar to the considerations for analyses of interview data.

The main objective of the study is to examine associations between WIC participation across the first five years of life and early childhood diet and health-related outcomes. In addition, the study also investigates other constructs that are closely tied to nutrition and health, including household food security status, caregivers' feeding practices and beliefs, and children's feeding environments.

To achieve these objectives, a representative sample of WIC enrollees was needed. The sample was intended to represent the national population of infants enrolled in WIC for the first time either while the mother is pregnant, or postnatally before 3 months of age, whose mothers are at least 16 years old and speak either English or Spanish; however, for operational reasons, exclusions based on geography and site size (discussed below) were necessary. In generalizing study findings to the population of new WIC enrollees, the implicit assumption is that the characteristics of the portion of the population not covered due to these restrictions and exclusions are the same as the characteristics of the portion of the population that is covered. Because the recruitment period for the study spanned 20 weeks, estimates from the study represent infants in the represented population who enrolled during that 20-week period (July–November 2013), rather than the monthly or annualized total number of WIC participants nationally.

Because a prenatal interview or a postnatal interview soon after birth was required, it was necessary to sample and enroll eligible infants into the study soon after WIC enrollment, but a national frame of WIC enrollees was not available. Therefore, we decided to use a TLS approach where we first sampled WIC sites, and then sampled eligible women who visited those sites to enroll their children into WIC during prespecified, randomly selected recruitment windows. The sites were sampled using a stratified two-stage sample design, and the recruitment windows were established with the goal of identifying approximately the same number of new WIC enrollees during an assigned recruitment window in each of the selected sites. In sampling the new WIC enrollees, we designed the sample to oversample those with certain characteristics (defined by combinations of the mother's race, ethnicity, trimester at WIC enrollment, and pre-pregnancy body mass index (BMI)) to increase the precision of estimates for subgroups of interest.

There are many considerations that play into the design of a stratified two-stage sample, including the development of the strata, the overall target sample size, the number of first-stage units to be selected, and the allocation of the sample to the strata. Likewise, there are many aspects to be addressed in the development of the weighting methodology for a study like this. Decisions involving these aspects of the sample design and weighting are beyond the scope of this manuscript; here, the focus is instead on the implementation of the sample design and the specific effects of TLS on weighting. Detailed discussion of the sample design and weighting may be found in Borger et al. (2022) [10].

## Sampling of WIC sites

At the time sampling began in 2012, there was no national list of all WIC sites. In order to sample WIC sites, we used data from the WIC Participant and Program Characteristics 2010 (PC 2010; see https://www.fns.usda.gov/wic/wic-participant-and-program-characteristics-2010) to

construct a sampling frame. State and local WIC agencies review applicant and participant data to determine ("certify") eligibility for benefits and to issue food vouchers and checks. State agencies routinely submit summaries of a minimum data set (MDS) of those data for a census of WIC participants, in this case, the PC 2010.

The PC 2010 data were summarized at the unit level reported by each state agency in the census of April 2010 (i.e., reporting units); as we will describe later, we obtained updated enrollment counts as of January 2012 for the purposes of determining sites' eligibility and selecting sites, and obtained updated enrollment counts as of July 2012 for the purpose of determining the length of the recruitment window. Due to flexibility in reporting unit level permitted for the submission of the census data, the PC 2010 file contained a mix of two unit levels–individual sites and local agencies. The sites are the individual locations that provide services to WIC clients, and the local agencies are organizations that provide administrative support and oversight to the sites. Because of this variation in reporting unit levels, we could not construct a frame of individual sites directly from the PC 2010. Rather than requesting a list of individual sites for all local agencies with no site information in the PC 2010, we used a two-stage sample design with two phases of selection in the first stage to ultimately get to a sample of sites (see Fig 1).

Because PC 2010 data were from April 2010, and thus were two years old by the time of site sampling for the WIC ITFPS-2, they did not provide information about the current eligibility of each reporting unit. Thus, we selected a larger initial sample of reporting units and subsampled eligible units to generate the first-stage sample of reporting units.

In the first phase of stage one of sampling (i.e., the selection of sites), we first created a reporting unit measure of size using the estimated number of eligible new WIC enrollees based on the PC 2010 data. Because the study recruitment period was limited, reporting units with estimated new WIC enrollment of less than 30 eligible women per month (assuming 20 WIC enrollment days per month) were considered ineligible. In light of cost and resource considerations, reporting units in American Samoa, Guam, Northern Mariana Islands, and US Virgin Islands were classified as geographically ineligible. With an objective of ultimately sampling 80 sites, the remaining reporting units were stratified into 40 strata using a combination of state agency policies and characteristics, and reporting unit characteristics from the PC 2010. We then sampled four reporting units per stratum with probabilities proportional to the measure of size.

In order to determine the eligibility for the phase 2 selection of reporting units and for the second stage of sampling, we requested information from state agencies for each reporting unit selected into the phase 1 sample. This included more recent WIC enrollment information from January 2012 (the most recent month available at the time of this exercise) and, for reporting units that were local agencies, a list of sites associated with the local agency. Sites expected to be non-operational for the next 12 months and those with low WIC enrollment were considered ineligible. If a local agency was selected in phase 1, the unit was eligible for phase 2 sampling as long as at least one site associated with that agency was eligible. After determining the eligibility of each of the 160 reporting units selected in phase 1, we then subsampled 2 reporting units per stratum with equal probabilities, resulting in 80 sampled reporting units at the end of the first stage of sampling.

In the second stage of sampling, for reporting units selected in the first stage that were local agencies, we selected one site per agency with probability proportional to the site's measure of size (PPS). This resulted in a sample of 80 sites. In addition to the original sample of 80 sites, we assigned replacement sites for each sampled site in case of site nonresponse (non-cooperation). Replacement sites were selected from the same stratum as the original sampled site such that the replacement site had a measure of size closest to that of the original sampled site.

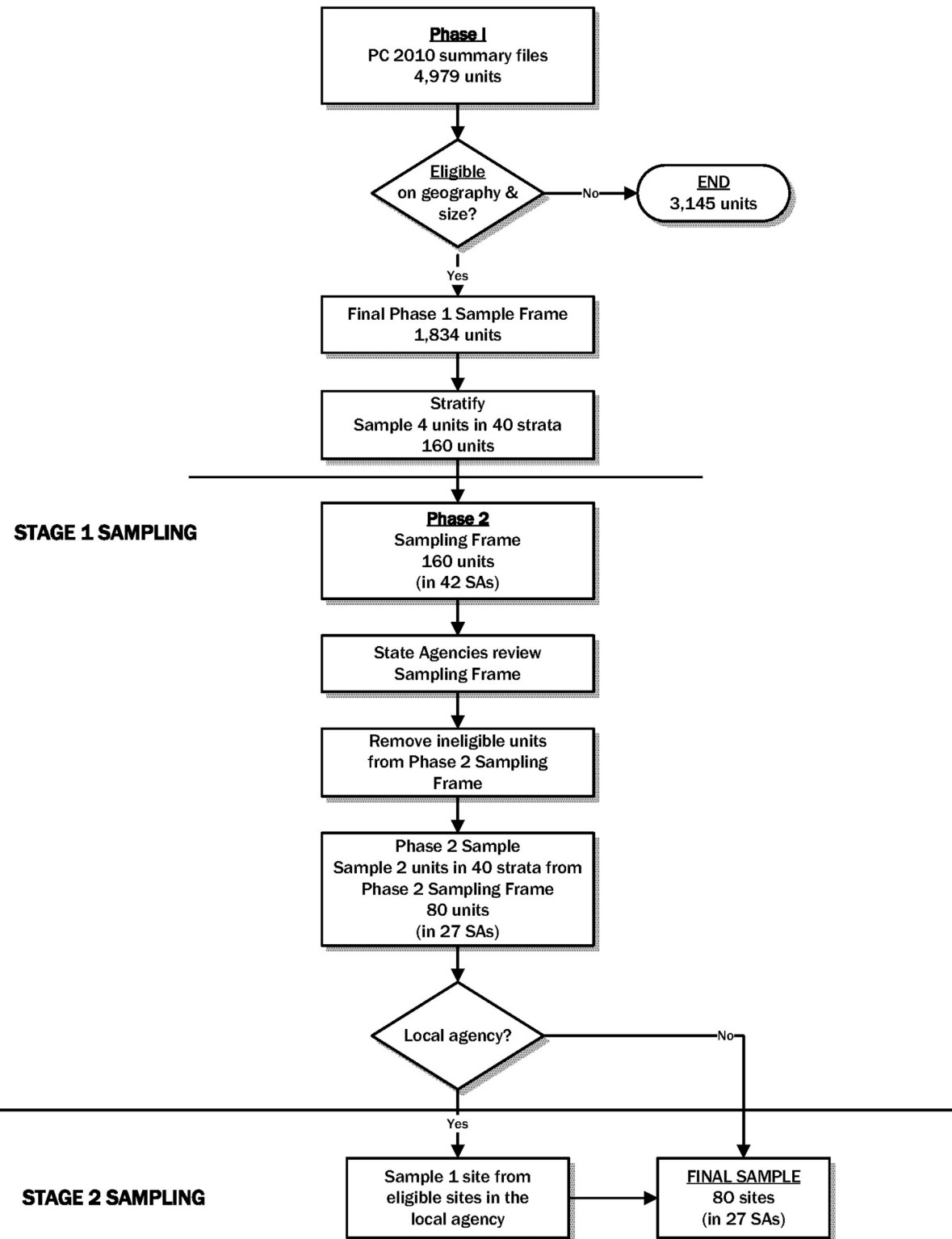

**Fig 1. Overview of WIC ITFPS-2 site sampling process.**

## Sampling of WIC enrollees

Interviews at select months were designated as "key" interviews, with a specified level of precision targeted for subgroup analyses from those interviews. For other interviews, a lower level of precision was deemed acceptable. To support the subgroup precision requirements for the key interviews, an oversample of participants with certain characteristics was desired. To facilitate the two levels of precision, we designated two samples, a core sample and a supplemental sample, with the distinction that the core sample would receive all interviews, while the supplemental sample would receive only the key interviews (see S1 Fig). The supplemental sample was thus designed, when combined with the core sample, to provide the oversamples needed to support subgroup analyses from the key interviews. While the core sample could be analyzed alone, the supplemental sample alone was not representative; it was only designed to be analyzed in combination with the core sample.

If a list of new WIC enrollees (along with their characteristics) had been available for use in sampling, then we could have sampled new WIC enrollees differentially based on these characteristics. Because such a list was not available, during recruitment on site, new WIC enrollees were screened for their eligibility and were asked to provide characteristics required in subsampling, which included the mother's race, ethnicity, trimester at WIC enrollment, and pre-pregnancy body mass index (BMI). Initially, half of new WIC enrollees were assigned to the core sample and the other half to the supplemental sample at each site. The subsampling rate of one-half is because meeting the precision requirements for the rarest subgroups required essentially double the sample size that was required for the more prevalent subgroups (those supported by the core sample). All those assigned to the core sample were retained in the sample, but those assigned to the supplemental sample were subject to subsampling with rates determined based on estimated population distributions, to meet the precision requirements. The subsampling was implemented using an algorithm that ran within the study screening instrument. The sampling plan described here was implemented in the early weeks of study recruitment.

The PPS selection of sites was designed to result in an approximately equal probability sample of study participants while allowing for a workload (in terms of number of enrolled study participants) that was balanced across sites. Specifically, we aimed to sample approximately 98 new WIC enrollees at each site. The approach we used was to assign in advance a "recruitment window"—a block of time during which study recruiters would attempt to recruit (or screen out) all new WIC enrollees at the site.

To assign the recruitment window for a site, we first needed to determine how long the window had to be in order to expect to approach for recruitment the target of 98 women at the site. Naturally, the WIC enrollment rate and schedule varied from one site to another, and we needed to account for these variations.

Although we had WIC enrollment information obtained from state agencies for January 2012, since WIC enrollment flows could change over time, prior to assigning the recruitment windows for each site, we gathered updated WIC enrollment information for July 2012 from the participating sites. Specifically, for each site, we obtained the monthly counts of prenatal WIC enrollees and postnatal WIC enrollees, and information on the days a site was open for new WIC enrollment. Because only prenatal enrollees and infants less than 2.5 months old being enrolled for the first time were eligible for the WIC ITFPS-2, we estimated the average daily WIC enrollment of eligible participants as the sum of the number of prenatal WIC enrollees and 20% of postnatal WIC enrollees (both from July 2012), divided by the number of WIC enrollment days scheduled for July 2012. This average daily enrollment was then used to estimate the number of recruitment days needed to achieve 98 women approached for recruitment at each site.

It is important to note that the recruitment windows determined based on the aforementioned average daily WIC enrollment calculations may result in fewer or greater than 98 study women approached for recruitment as a result of day-to-day fluctuations or trends in WIC enrollment at the site (including fluctuations due to seasonal variation in births, e.g., [11]). If there was a significant change in the WIC enrollment flow between July 2012 and the actual recruitment period (July through November 2013), a site with lower than estimated flow would have fewer women than targeted to approach for recruitment, whereas a site with higher than estimated flow would have more women than targeted.

Additionally, the actual daily WIC enrollment flow could vary considerably from the average daily WIC enrollment depending on other activities planned at the site. For example, some sites had specific days of the week that typically had more prenatal WIC enrollees because of the classes offered at that site on those days. There were also sites that scheduled advance appointments on certain days of the week, and walk-ins on the rest. Recruitment window lengths were typically longer than a week, so day-to-day WIC enrollment fluctuations within a given week would not be likely to have large impacts on study enrollment, but week-to-week or seasonal fluctuations could result in systematic shortfalls or surpluses in study enrollment. Another factor that could affect the WIC enrollment flow (and, thus, study enrollment) is an event such as government sequester or a natural disaster. With a relatively lengthy study recruitment period, there was an increased chance that such events could unexpectedly cause a change in the WIC enrollment flow.

After determining the number of study recruitment days required for each site, we then needed to determine which days within the recruitment period each site would be accepting new WIC enrollments so that we could determine the lengths of the site recruitment windows. A naïve approach might have been to assume that all sites enrolled new WIC participants every weekday (Monday through Friday). However, in many cases, this assumption would have resulted in window lengths that were too large or too small because the actual WIC enrollment schedules turned out to vary from site to site. While many sites enrolled new WIC participants on every weekday, there were sites that were open only on certain days of the week, sites that enrolled new participants only on certain days (not all days they were open), and sites that were open and enrolling new participants on some Saturdays in addition to weekdays. Obtaining accurate information about the site's WIC enrollment schedule required clear communication with the sites to confirm that all enrollment days were included. For example, if a site used advance appointments on most days, but allowed walk-in enrollments on other days, the WIC enrollment schedule needed to include both appointment days and walk-in days.

Besides the regular WIC enrollment schedule, it was also necessary to consider site closures due to holidays (e.g., Labor Day), staff training days, and other site-specific reasons. For example, some sites were located within facilities shared with other organizations and had to follow the other organizations' closing schedules. We collected information on planned closures during the study recruitment period (July through November 2013) to account for this when determining the site's recruitment window.

Once the sites' recruitment window lengths had been determined, in order to generate a random site recruitment window within the study recruitment period, we needed to assign each site its recruitment start date, but there were two requirements to consider. The first consideration was a requirement to stagger the start of recruitment for sites over nine weeks in order to efficiently allocate study resources. The second requirement was to randomly generate the assignments. Random assignment was key to ensuring a probability sample that appropriately captured the variation in "busyness" among the sites. This was done both to help ensure that the overall target sample size was met (as some sites may have enrolled more and the

others less than expected, depending on how busy or slow they were), and in case the experiences differed for WIC participants if they visited a site during a busy period versus a slower period.

In assigning the site recruitment start dates, it was necessary to first determine the last day in the study recruitment period that each site could begin study recruitment and still be allotted the required number of recruitment days, based on their window lengths and WIC enrollment schedules. For example, if a site required 14 weeks to complete the recruitment, this site's start date could not be in the last release group (which was designated to start recruitment with only 9 weeks and 5 days remaining in the study recruitment period). On the other hand, if a site only required 2 weeks for recruitment, they were eligible for any release groups including the group released with approximately only 10 weeks remaining.

Then, we randomly selected sites into nine release groups (one group released each week) taking into account the eligibility of each site to be in a given release group. To allow for ramping up and ramping down of the study recruitment effort, the first and the last release groups were designed to contain 5 sites each, and the remaining release groups contained 10 sites each.

All prenatal enrollees and postnatal enrollees less than 2.5 months old who enrolled in WIC for the first time during the recruitment windows at the selected sites were eligible for the study, regardless of the mother's prior enrollment in WIC for a previous pregnancy or child. With time-location sampling, in order to properly compute probabilities of selection, it is essential to know where and how often a given individual could have been selected, which requires knowing where and how often the individual visited eligible venues during the recruitment period. For example, someone who frequently visits a given venue has a higher probability of selection (all other things being equal) than those who only visit occasionally. In our study, however, because only those enrolling in WIC for the first time on the particular visit were eligible, there was no issue of multiple chances of selection. Note that participants can only enroll in WIC at one site within one state agency. In the case of multiple births (e.g., twins), we sampled exactly one among the multiple births and accounted for this special sampling step in the computation of the probabilities of selection.

Selected sites varied in their schedules and hours of operation, use of advance appointments, average enrollment of new WIC clients, percentages of WIC enrollees who only spoke Spanish, and reliability of cell signal or Wi-Fi or Ethernet availability. Because of these variations, a recruitment procedure suitable for some sites may not be the best for the others. Therefore, recruitment models for individual sites were coordinated depending on these characteristics.

Two general recruitment approaches were used. Sites were assigned to either an on-site recruitment model or a referral-form-only recruitment model. In both models, referral forms were filled out by the WIC site staff with basic information about all women believed to be eligible for the study. In the on-site recruitment model, the completed referral forms were given to the study recruiter, and the study enrollment process continued on-site. In the referral-form-only model, the completed hard-copy referral forms were given to the study recruiter, but the recruiter followed up with the WIC enrollees via phone at a later time.

In either model, because the WIC site staff filled out the referral forms rather than the study recruiter taking on the recruitment process from the beginning, there was a possibility that some eligible WIC enrollees might have been missed or lost in this step. The possibility of failing to identify persons eligible for the study is an important aspect to consider in developing and designing study recruitment and enrollment procedures, and researchers must minimize such possibilities in order to limit bias. We took several steps to minimize the non-identification of eligible persons. First, prior to the start of recruitment at a site, we provided WIC staff

with training on identifying study-eligible participants. Second, WIC staff were instructed to complete a referral form for each study-eligible participant (without contact information, for those who did not consent), regardless of whether they consented to be contacted for the study, and those forms were provided to the recruiter for the purpose of accounting for and adjusting for failure to obtain consent. Third, with the on-site recruitment model, during their downtime and with the site's permission, the recruiters also monitored the waiting area to assess the flow. Fourth, if the flow rate was significantly lower than expected, we asked the site to review their records for any potentially missed eligible participants. Fifth, in a few cases, study supervisors visited the site in order to monitor the flow in person.

Between the two recruitment models, the preferred approach was to send a recruiter to complete the study enrollment process on site. Doing so allowed for an immediate attempt to recruit eligible WIC enrollees in-person, and both the immediacy and the in-person aspect were expected to result in higher response rates than the referral-form-only recruitment model by phone. However, in some sites, this approach was not feasible or it was deemed more efficient to use the referral-form-only recruitment model. The determination of which model to use was based in part on the availability and reliability of cell phone signals and internet as they were essential in launching the interview instrument, as well as estimated WIC enrollment rate and the size of the space available for study recruitment.

For the sites with the on-site recruitment model, the site's enrollment schedule and hours of operation, expected flow rate, and use of advance appointments for new WIC enrollments were taken into consideration in planning when and how many recruiters to send. While the majority of sites scheduled WIC enrollment appointments in advance, the proportions of advance appointment enrollments and walk-in enrollments varied among and within the sites. Some sites took exclusively walk-in enrollments. With advance appointments, the schedules were shared with study recruiters one or two days in advance, which allowed the recruiters to arrive at the right time and reduced time spent at the site without any new WIC enrollees.

Another consideration for staffing of the study recruitment was the prevalence of WIC enrollees at the site who spoke Spanish only. Bilingual recruiters were assigned to sites where more than 15% of WIC enrollees were estimated to speak only Spanish. In other sites, English-speaking recruiters were assigned, and if there were Spanish speaking participants, recruiters contacted their bilingual study liaison who conducted the study enrollment process with the participants via phone.

## Sample monitoring and design modification

As noted earlier, each site was assigned a recruitment window based on the expected daily WIC enrollment at the site and the site enrollment schedule. As added insurance, we had inflated the recruitment window lengths for all sites by 3 percent (in terms of number of recruitment days) prior to the start of the recruitment effort due to a concern about declining WIC enrollment. However, as we monitored the recruitment progress for a few weeks, we determined that the projected yield was expected to be lower than initially estimated. Therefore, in an effort to achieve the desired sample size while retaining the statistical integrity of the sample, we extended recruitment windows for all sites. Taking into account the new projected yield, additional burden on the sites, and the resulting impact on the length of the entire study recruitment period, the recruitment window for each site was extended by 10 percent. Also, the recruitment windows had to be adjusted as we learned that the site-specific WIC enrollment and closure schedules were different from what we knew prior to the window assignment.

As we continued to monitor the yield for several weeks, we noticed that the overall WIC enrollment was still too low so that a further adjustment was necessary to meet the target sample size even after this extension of the recruitment windows. Additionally, the prenatal/postnatal enrollment distribution of the sample was skewed more toward prenatal enrollees than expected. To address these concerns, we changed the process of participant sampling to remove the subsampling of the supplemental sample.

### Implications of TLS for the computation of survey weights for analysis purposes

In this study, to allow for representative analyses, survey weights that account for the unequal selection probabilities are needed. The weights account for the probability of selecting the sites, adjust for the proportion of new enrollees within the study recruitment period who enrolled during the particular set of recruitment days for the site (accounting for the adjustments to the lengths of the recruitment windows), and account for the subsampling of participants into the core or supplemental samples including applicable modifications.

As noted above, the length of the recruitment window was based on the expected daily WIC enrollment at the site and the site enrollment schedule. Let $L_h$ denote the length of the recruitment window for site $h$, and let $L$ denote the length of the study recruitment period. Then the proportion of new enrollees within the study recruitment period who enrolled during the site recruitment window is estimated by $L_h/L$. All adjustments to the recruitment windows are reflected in these quantities; i.e., $L$ and $L_h$ are the final study and site recruitment window lengths, respectively.

The computation of the weights begins with the reciprocal of the probability of selection of the reporting unit (which takes into account both phases of selection). If the sampled reporting unit was a local agency, this weight is multiplied by the reciprocal of the probability of selection of one eligible site among the eligible sites associated with the reporting unit. The result is a site-level weight. The site-level weight is then multiplied by the factor $L/L_h$ to account for the proportion of new enrollees enrolling during the site recruitment window. The result is a weight that is initially applied to all new enrollees approached by the study. This enrollee-level weight is then adjusted for screener nonresponse, for subsampling in the selection of the supplemental sample (using the actual subsampling rates applied at the time of screening; following the elimination of all subsampling, this factor was set equal to 1), for failure to obtain consent to participate in the study, and for interview nonresponse.

Weights for analyses of interview data were created for the core sample only or the combined core and supplemental samples, depending on which sample(s) were administered the particular interview(s). Details of the computation of the weights may be found in Borger et al. (2022) [10].

### Results

Among 4,979 reporting units in the PC 2010, 17 units in American Samoa, Guam, Northern Mariana Islands, and US Virgin Islands were considered ineligible due to geographical reasons, and an additional 3,128 units were ineligible due to low expected WIC enrollment, leaving 1,834 eligible units (36.8% of all reporting units; the excluded low-enrollment sites are estimated to comprise about 13 percent of total new WIC enrollments). In the phase 1 of the first stage sampling of sites, 160 reporting units (4 units per stratum in each of 40 strata) were selected from the 1,834 units. Based on updated information obtained from the 42 state agencies associated with the 160 units, eligibility of the 160 phase 1 sampled units was determined,

and in phase 2 of the first stage of sampling, 80 units (2 units per stratum) were selected among the eligible units. For each unit selected in phase 2, if the unit was a local agency, then eligible sites associated with that local agency were enumerated, and one site per local agency was selected in the second stage of sampling. This resulted in a final sample of 80 sites associated with 27 state agencies. Six (7.5%) of the originally sampled sites did not agree to cooperate with the study and were thus replaced by their predesignated replacement sites selected from the same stratum with similar measures of size.

Once the cooperating sites had been determined, a recruitment window was assigned for each site. Based on the number of days expected to be needed to identify 98 women to approach for recruitment and the WIC enrollment schedule at each site, window lengths were calculated. The number of recruitment days needed per site ranged from 4 to 87 days after all adjustments, and the calendar length of the recruitment window varied from 2 weeks to nearly 18 weeks with the majority (54%) of the sites requiring 3–8 weeks (see Fig 2). While the majority of the sites enrolled women in WIC on every weekday, 33 out of 80 sites had WIC enrollment schedules that were not Monday-Friday.

At each site, the number of recruiters and the recruitment model were determined based on site characteristics. Of the 80 sites, 42 sites were open for new WIC enrollments more than 40 hours a week, and 13 sites enrolled 50 percent or fewer of WIC participants through advance appointments. A total of 11 sites required more than one recruiter on site due to the number of hours they were open for enrollment and the proportion of enrollment done via walk-ins. Because of the WIC enrollment flow, space limitations at the site, and/or cell phone use limitations, 5 sites were assigned the referral-form-only recruitment model. Among the 75 sites that used the on-site recruitment model, 46 sites estimated more than 15% of the WIC enrollees to only speak Spanish, and therefore bilingual recruiters were assigned to those sites.

Target and actual recruitment and enrollment sample sizes are shown in Table 1. The initial sample design targeted 7,840 women approached for recruitment (i.e., 98 per site for each of 80 sites); the actual number of women approached for recruitment was 6,775. The target sample sizes were 4,435 women enrolled in the study and 3,990 live births to women enrolled in the study. The majority of the difference between the number approached for recruitment and the number enrolled in the study was by design, due to subsampling of women assigned to the supplemental sample. Of 3,990 study enrolled live births, 2,804 or approximately 70% were expected to be in the core sample and 1,186 or approximately 30% were expected to be in the supplemental sample. In the actual sample, because of the change in the sampling process to adjust for the declining WIC enrollment and a higher than expected proportion of prenatal WIC enrollees, we enrolled 4,367 women in the study, out of 6,775 women approached for recruitment. Of the 4,367 women enrolled in the study, 3,732 were prenatal enrollees; the study included 3,411 live births to women enrolled in the study prenatally. Among woman/infant dyads enrolled in the study, there were 4,046 actual live births, compared to the targeted 3,990. Among the 4,046 enrolled live births, 3,235 (approximately 80% of those enrolled) were assigned to the core sample and 811 were assigned to the supplemental sample. Although the resulting distribution of core and supplemental samples differed from the original plan, the adjustments to the recruitment windows and to the supplemental sample subsampling plan successfully achieved and exceeded the target sample sizes for the core sample and for overall enrolled live births. For subgroups, the deviations from the expected sample sizes were larger, as shown in Table 2. In particular, there were shortfalls of enrolled women identified as Black or other race and of WIC participants who enrolled in WIC postnatally.

There was considerable site-to-site variation in the number of new WIC enrollees approached for recruitment. Fig 3 is a scatterplot of the number of women approached for recruitment by the number of recruitment days at each site. The symbols used in this graph

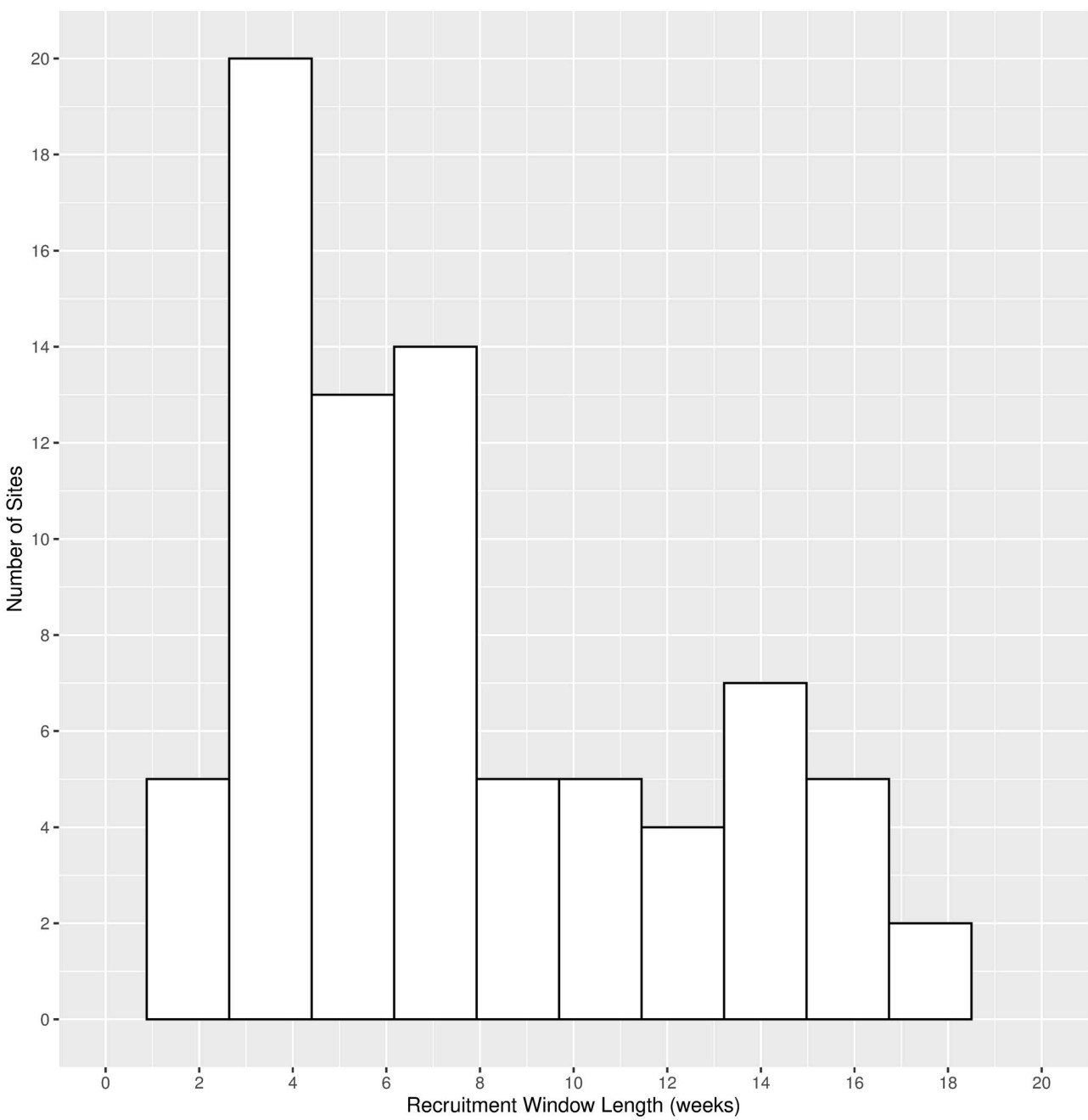

**Fig 2. Recruitment window lengths in weeks.**

**Table 1. Recruitment and enrollment sample sizes: Target vs. actual.**

|                          | Target | Actual |
|--------------------------|--------|--------|
| **Approached for recruitment** | 7840   | 6775   |
| **All enrolled**         | 4435   | 4367   |
| **Enrolled—live births** | 3990   | 4046   |

**Table 2. Ratio of actual to expected enrolled sample size, by subgroup.**

| Subgroup | Number Enrolled: Ratio of Actual to Expected Sample Size |
|---|---|
| Overall | 1.0 |
| Mother's race | |
| Black | 0.8 |
| White | 1.7 |
| Other | 0.4 |
| Mother's Hispanic origin | |
| Hispanic | 1.2 |
| Non-Hispanic | 0.9 |
| Timing of WIC enrollment | |
| 1st trimester | 0.8 |
| 2nd or 3rd trimester | 1.6 |
| Postnatal | 0.5 |
| Pre-pregnancy body mass index (BMI) of mother | |
| Overweight | 1.1 |
| Obese | 1.0 |
| Other | 0.9 |

distinguish between sites where the entire sampling was done using the original subsampling approach (triangles), sites with no subsampling (circles), and sites with subsampling for some time during their recruitment window but not the whole window (plus signs). While most sites had a mixture of subsampling, and there were only a few sites which had subsampling for the whole recruitment period, we can see that those with the original subsampling generally had fewer women approached than the target of 98 (dotted line). This is consistent with the observation made during the recruitment that the WIC enrollment flow was lower than expected, which contributed to the decision to drop the subsampling of supplemental sample.

There appears to be a slightly positive relationship between the number of women approached for recruitment and the number of recruitment days. This is likely because the day-to-day and week-to-week variability of WIC enrollment flow was not covered well for shorter recruitment groups, whereas the longer recruitment groups had more opportunities to balance out shortfalls with higher-recruitment days/periods.

The scatterplot also shows an outlier, a site with 264 women approached for recruitment during 65 recruitment days. This site anticipated a lower-than-normal WIC enrollment flow during the last week of July through the second week of August due to preparation for schools starting, and expected a higher flow afterwards. The expected WIC enrollment rate could have been underestimated because it was based on data from July 2012, which included a portion of the "low" period. This site had a long recruitment window (13 weeks), extending to October, and therefore had more time to make up for any "loss" during the low enrollment period. The combination of the enrollment flow estimate having been based on a lower-than-average flow period and the site's recruitment period covering a higher-than-average flow period likely combined to result in the unusually high number of women approached for recruitment at this site.

## Discussion

Time-location sampling is a useful approach for obtaining a sample in situations in which no list of population members is available and a substantial proportion of members of the population visit specific types of locations. While TLS has generally been used in studies of stigmatized populations or those engaging in illegal behaviors (such as illicit drug users, men who

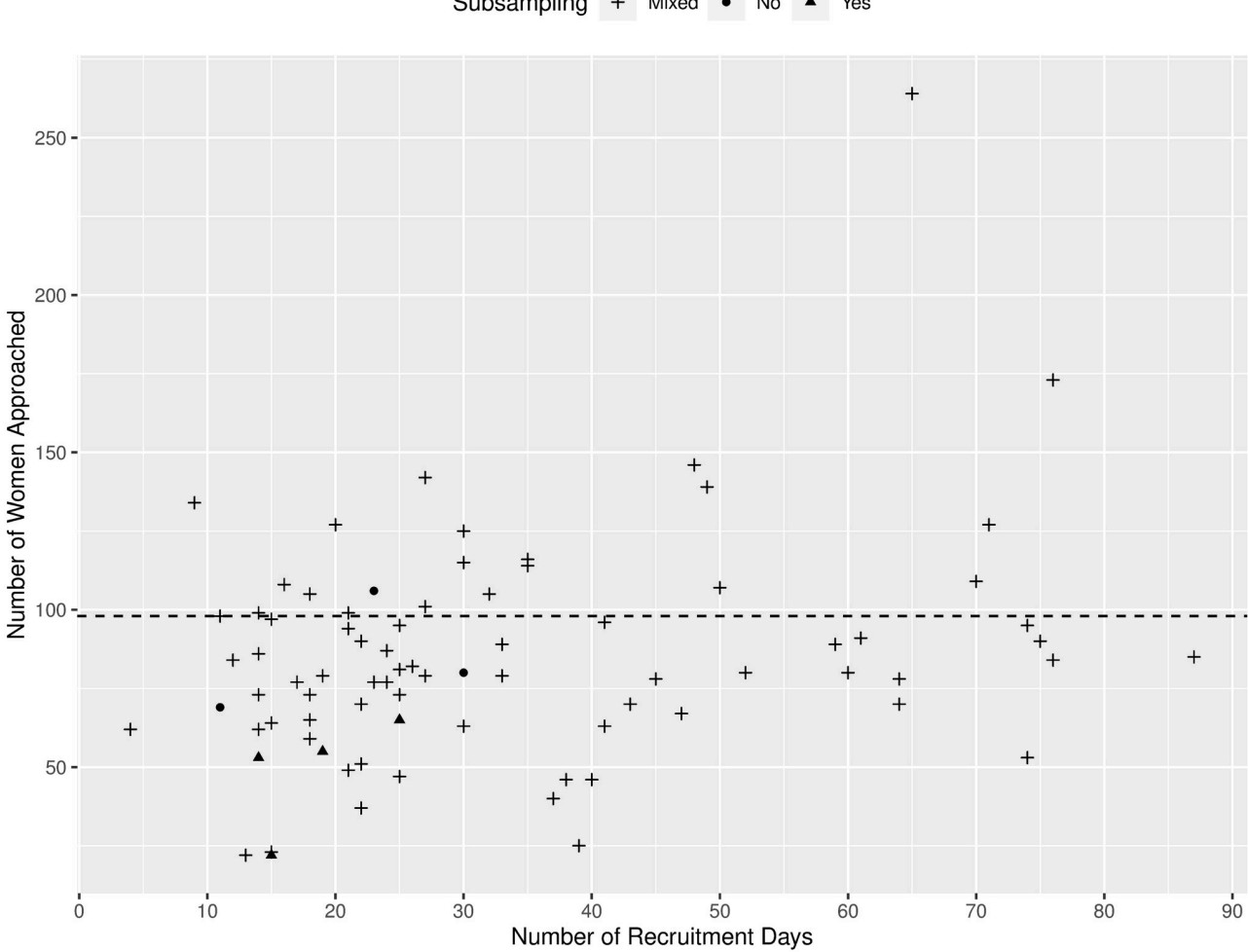

**Fig 3. Number of women approached for recruitment at the site vs. number of recruitment days.** Note: the "Yes" symbol indicates that subsampling for the supplemental sample was conducted throughout the recruitment period in the site; "No" indicates that the site's recruitment period began after subsampling for the supplemental sample had been stopped; and "Mixed" indicates that the site's recruitment period began with subsampling, but the change to stop subsampling was made during the recruitment period in the site.

have sex with men, and female sex workers), the application discussed here demonstrates its utility in selecting samples of program participants such as WIC enrollees. Additionally, as we have demonstrated through our application, there are situations in which a carefully designed TLS may be used to produce a probability sample.

We have described various issues and considerations that pertain to TLS, and used our application to illuminate aspects of these. These issues and considerations include

- Identifying an appropriate type (or types) of locations frequented by members of the target population;

- Gathering enrollment count information at each location for use in PPS selection of locations;

- Determining locations' days/hours of operation (as they pertain to population members' visits) for the purposes of establishing study recruitment window lengths for sampled locations;

- Randomizing the assignment of the recruitment windows to balance across heavier and lighter flow periods and, where applicable, account for temporal differences;

- Staffing the locations adequately to manage the within-site sampling and recruitment activities; and

- Capturing the information necessary to compute survey weights, such as individuals' frequency of visits to eligible locations.

With TLS, the study must be prepared to handle location-to-location variations in sample yield that will undoubtedly occur due to deviations from expected flow rates. These variations may impact both staffing needs and sample yield, so having the foresight and flexibility to address them is crucial.

In our implementation of the sample design for the WIC ITFPS-2, we found there was a greater decline in WIC enrollment than expected. Therefore, we extended the recruitment periods by a fixed percentage just prior to the start of recruitment as well as during recruitment. For a future study, it may be better to build in an extra step at the sample design stage to carefully examine recent enrollment trends and adjust the recruitment period lengths accordingly. We also found that some sites had large differences between the expected and the actual enrollment flows during recruitment. If resources are available, it may help to consider enrollment data over more months (e.g., for each of the past 12 months) in order to take into account any month-to-month variation in estimating the enrollment flow. However, since such information will need to be provided by the sites, site resources must be considered to determine if this is feasible and practical.

## Supporting information

**S1 Fig. Overview of WIC ITFPS-2 interviews administered to subsamples.**
(DOCX)

## Acknowledgments

For their feedback on a draft of the manuscript, the authors express their appreciation to Christine Borger, Mike Brick, Janice Machado, Courtney Paolicelli, Kelley Scanlon, and Ting Yan. We also greatly appreciate the feedback and suggestions provided by the Guest Editor (Paul A. Smith), the original associate editor, and three reviewers. All of this feedback resulted in substantial improvements to the manuscript.

## Author Contributions

**Methodology:** Jill M. DeMatteis.

**Visualization:** Yumiko Siegfried.

**Writing – original draft:** Yumiko Siegfried, Jill M. DeMatteis.

**Writing – review & editing:** Yumiko Siegfried, Jill M. DeMatteis, Bibi Gollapudi.

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
