## [Decision Letter · Decision Letter 0]

28 Sep 2022

PONE-D-21-33776Designing a sample of program participants using time-location samplingPLOS ONE

Dear Dr. Siegfried,

Thank you for submitting your manuscript to PLOS ONE. After careful consideration, we feel that it has merit but does not fully meet PLOS ONE’s publication criteria as it currently stands. Therefore, we invite you to submit a revised version of the manuscript that addresses the points raised during the review process.

 Your manuscript has been assessed by three peer-reviewers and their reports are appended below.  The reviewers comment that several aspects of the manuscript need further information and/or clarification, especially the aspects of the manuscript pertaining to the process of the survey design. In addition, the reviewers have raised some questions with the statistical analyses reported in the study.  Could you please revise the manuscript to carefully address the concerns raised?

We look forward to receiving your revised manuscript.

Kind regards,

Maria Elisabeth Johanna Zalm, Ph.D

Editorial office

PLOS ONE

Journal Requirements:

Reviewers' comments:

Reviewer's Responses to Questions

**Comments to the Author**

1. Is the manuscript technically sound, and do the data support the conclusions?

Reviewer #1: Yes

Reviewer #2: Partly

Reviewer #3: Yes

2. Has the statistical analysis been performed appropriately and rigorously? 

Reviewer #1: Yes

Reviewer #2: No

Reviewer #3: Yes

3. Have the authors made all data underlying the findings in their manuscript fully available?

Reviewer #1: Yes

Reviewer #2: No

Reviewer #3: Yes

4. Is the manuscript presented in an intelligible fashion and written in standard English?

Reviewer #1: Yes

Reviewer #2: Yes

Reviewer #3: Yes

5. Review Comments to the Author

Reviewer #1: I had the pleasure of reviewing this manuscript and recommend that it be accepted without major revisions. The authors clearly stated the need for this manuscript, detailed their method by describing an example application, and provided a reasonable discussion. Aside from minor stylistic choices the authors might revisit before publication, depending on the journal's style (e.g., using "data" in the singular - line 167, or capitalizing State Agencies while keeping local agencies in lower case), I have no recommendations for these authors. Excellent work.

Reviewer #2: See attached file.

(the remainder of the text is just to meet the minimum character count so that the review can be submitted through an intransigent system that insists that this box contains at least 200 characters)

Reviewer #3: The manuscript “Designing a sample of program participants using time-location sampling” details the approach developed to use time-location sampling (TLS) in 2013 to enroll a cohort for the WIC Infant and Toddler Feeding Practices Study-2. This complex sample design needed to account for numerous considerations, including identifying the appropriate types of locations, estimating flow rates, determining days/hours of operation to establish study recruitment lengths, randomizing the assignment of recruitment windows to balance different flow periods, adequate staffing, and calculating appropriate survey weights. This carefully designed time-location sample enabled a probability sample to be produced. The manuscript is well-written in easy-to-understand language and details the approach, challenges and solutions, and limitations of this study. Because TLS generally produces non-probability samples, the strategy detailed here may be helpful to others looking to implement a similar approach. To that end, I have some comments and suggestions provided below to help make the paper more generalizable.

Major Comments:

1. There are only four citations, and only one of those is directly related to TLS. I would suggest extending the literature review in Section 1, perhaps highlighting in particular other examples of where TLS was used to create a probability sample.

2. How were the various sample sizes determined? Why 80 sites (line 177)? Why 98 women per site (line 222)? Information on how these and other sample size numbers were obtained (and why) would be helpful to other practitioners. At least provide a reference and justification if using a standard formula.

3. The “Computation of survey weights for analysis purposes” section needs more detail. At minimum, a formula and/or citation should be provided. What would be even more beneficial and strengthen the work is to provide some detail about how the survey weights were actually calculated since you repeatedly stress that what makes this survey unique is creating a probability sample from TLS. That makes it seem like figuring out the survey weights is non-trivial, particularly given the complexity of your design and adjustments needed to the procedure. Even if the goal of this paper is to lay out the approach and not focus on data analysis/final estimates, the approach is incomplete without knowing the weights. This would help make the work much more generalizable.

4. Lines 454-458: I am not sure that the regression line is the best way to demonstrate your point here. Two things to consider – 1) You mention the slope is positive (0.53), but don’t mention if this is actually statistically significantly different than 0 (i.e, do you reject the null hypothesis H0: beta1=0?). If not, even if the slope is positive, it could still just be due to chance. 2) You mention the outlier in the next paragraph. Is the slope still as positive (and/or still significantly different than 0) with the outlier removed? I think you could still have made your point without fitting the regression model, which adds these extra complexities/assumptions (e.g. considering case-influence statistics, if it’s appropriate to actually treat the sites as independent given the strata, etc.)

Minor Comments:

1. Line 148-150: It’s stated here that the sample was designed to “oversample those with certain characteristics to increase the precision of estimates for subgroups of interest.” Although these characteristics are mentioned later (if I’m interpreting correctly, they are mother’s race, ethnicity, trimester at WIC enrollment, pre-pregnancy BMI, household composition, and income), I think it would be helpful to mention them here.

2. I would recommend moving Figure S2 into the main text. The descriptions in the “Sampling of WIC sites” section were complex and having the figure there would be very helpful.

3. Relatedly, I’d recommend omitting figures 2 and 3, since they each provide little information since the pre and postnatal bars sum to 100% and the core and supplemental bars also sum to 100%. Those numbers/percents could be given in a table instead for easier comparison, as the exact number of live births is hard to read from the barplot.

4. The conclusion felt a little abrupt. Having a paragraph about lessons learned, or what could be done differently for a future study, would also help make the paper more generalizable.

6. PLOS authors have the option to publish the peer review history of their article (what does this mean?). If published, this will include your full peer review and any attached files.

Reviewer #1: No

Reviewer #2: No

Reviewer #3: No

---

## [Author Response · Author response to Decision Letter 0]

16 Feb 2023

We have revised the manuscript and related materials based on suggestions from the editor and reviewers. Please see "Response to Reviewers" document for more detailed response.

---

## [Editor Report · Decision Letter 1]

22 Mar 2023

PONE-D-21-33776R1Implementation of a sample design for a survey of program participants using time-location samplingPLOS ONE

Dear Dr. Siegfried,

Thank you for submitting your manuscript to PLOS ONE. After careful consideration, we feel that it has merit but does not fully meet PLOS ONE’s publication criteria as it currently stands. Therefore, we invite you to submit a revised version of the manuscript that addresses the points raised during the review process. Thank you for the careful revision of your paper. Following the first round of review (where I was one of the referees) I have taken over as Guest Editor for your manuscript, so I am now unmasked. The revised manuscript is substantially improved over the first version. Some further changes are, however needed.

The following issues must be addressed in a revision:

There is an important discrepancy between the responses to the reviewers and what the revised manuscript actually says. In the response to the reviewers you say of the exclusions of the smaller and geographically remote sites:

“the exclusions do not result in undercoverage error, because the study is not designed to cover and does not purport to cover enrollments at the low-flow or geographically ineligible sites. That is, the study population is defined with these exclusions duly noted, and there are no adjustments to the survey weights to represent these exclusions.”

But in the manuscript I read:

a representative sample of infants enrolled in WIC was needed (line 29)Our approach generated a probability sample (line 35-36)a national sample of Federal nutrition assistance program participants (line 83-84)our approach was designed to yield a probability sample of new enrolees (line 106-107)a representative sample of WIC enrollees (line 151)The sample was intended to represent the national population of infants enrolled in WIC for the first time either while the mother is pregnant, or postnatally before 3 months of age, whose mothers are at least 16 years old and speak either English or Spanish. (line 152-154)

These two points of view are not consistent. The manuscript needs to explain what part of the population is covered, and about which you can make representative or probability-based inferences. I don’t have a particular issue if you want to define the target population as all infants enrolled in WIC, but it needs to be clear that the sampling reaches only a smaller population, and that you must therefore assume that the characteristics of the part of the population that is not covered are the same as those of the part that is covered. This kind of approach is, after all, standard. But it needs to be clearly explained.

In the response to the referees you say “Removal of the subsampling in the supplemental sample results in increased sample sizes for subgroups that were previously being subsampled, thereby improving the precision of estimates for target subgroups.” But in fact you have achieved the required overall recruitment by removing subsampling, so the target subgroups (the ones that you intended the design to oversample) are no longer oversampled (except to the extent that they were before you changed the design). It is in fact the non-target subgroups which will receive larger sample sizes. Some assessment of the impact of this must be included.

In addition I have some further comments which you should consider in a revision:

line 227-8: Is the replacement unit similar in size because it was in the same stratum, or did you make them more similar in size than that? I note that with only 4 units per stratum the options for choice are rather limited (though there could be a wider choice where the unit is subsampled within a LA)line 260: Insert “(or screen out)” after “recruit”line 288: “are” -> “were”line 436: *L_h_*/*L* might be a reasonable approximation, but, as explained in my original review, it is not strictly true because the days do not have equal probabilities of inclusion in the sample (the days at the beginning and end of the study period are less likely to be included, unless you allow periods that "wrap" around to cover the beginning and the end, which is not very practical operationally). I think you need to explain this (and note that Borger *et al*. (2022) uses *L_h_*/*L* too).line 526-530: I have a different explanation/ suspicion about the estimates of flow rates, but your explanation is also possible. If it was just variability, I'd still expect a fit to be flat. But I think no further adjustment to the text is needed.You response to the referees about weights says that you added a reference to a report with details. I looked at the report and it is minimally sufficient; I note that the weighting is described only in words and no equations are given. You should still consider providing additional details about the weighting process in the paper, especially how it was adapted to changes made in the field, since that is the key topic in your paper. For example, certainty units are not mentioned.In Fig. 1 I don’t find it clear whether “160 units / 42 SAs” (and other similar) means that there are 202 things all together; I suspect not, but you could help with something like “160 units incl 42 SAs”.

We look forward to receiving your revised manuscript.

Kind regards,

Paul A. Smith

Guest Editor

PLOS ONE
---

## [Author Response · Author response to Decision Letter 1]

28 Apr 2023

Authors’ Responses to Reviewers’ Comments

Thank you for the careful revision of your paper. Following the first round of review (where I was one of the referees) I have taken over as Guest Editor for your manuscript, so I am now unmasked. The revised manuscript is substantially improved over the first version. Some further changes are, however needed.

The following issues must be addressed in a revision:

1. There is an important discrepancy between the responses to the reviewers and what the revised manuscript actually says. In the response to the reviewers you say of the exclusions of the smaller and geographically remote sites:

“the exclusions do not result in undercoverage error, because the study is not designed to cover and does not purport to cover enrollments at the low-flow or geographically ineligible sites. That is, the study population is defined with these exclusions duly noted, and there are no adjustments to the survey weights to represent these exclusions.”

But in the manuscript I read:

• a representative sample of infants enrolled in WIC was needed (line 29)

• Our approach generated a probability sample (line 35-36)

• a national sample of Federal nutrition assistance program participants (line 83-84)

• our approach was designed to yield a probability sample of new enrolees (line 106-107)

• a representative sample of WIC enrollees (line 151)

• The sample was intended to represent the national population of infants enrolled in WIC for the first time either while the mother is pregnant, or postnatally before 3 months of age, whose mothers are at least 16 years old and speak either English or Spanish. (line 152-154)

These two points of view are not consistent. The manuscript needs to explain what part of the population is covered, and about which you can make representative or probability-based inferences. I don’t have a particular issue if you want to define the target population as all infants enrolled in WIC, but it needs to be clear that the sampling reaches only a smaller population, and that you must therefore assume that the characteristics of the part of the population that is not covered are the same as those of the part that is covered. This kind of approach is, after all, standard. But it needs to be clearly explained.

In order to be explicit about the effects of the exclusions, we have made modifications to the manuscript in the places you indicated.

2. In the response to the referees you say “Removal of the subsampling in the supplemental sample results in increased sample sizes for subgroups that were previously being subsampled, thereby improving the precision of estimates for target subgroups.” But in fact you have achieved the required overall recruitment by removing subsampling, so the target subgroups (the ones that you intended the design to oversample) are no longer oversampled (except to the extent that they were before you changed the design). It is in fact the non-target subgroups which will receive larger sample sizes. Some assessment of the impact of this must be included.

Table 2 gives the ratios of the actual to expected enrolled sample sizes, by subgroup. 

In addition I have some further comments which you should consider in a revision:

a. line 227-8: Is the replacement unit similar in size because it was in the same stratum, or did you make them more similar in size than that? I note that with only 4 units per stratum the options for choice are rather limited (though there could be a wider choice where the unit is subsampled within a LA)

The replacement unit was the unit, within the same stratum as the original unit, that was closest in size to the original unit. We revised the wording to be more clear about this.

b. line 260: Insert “(or screen out)” after “recruit”

This change was made.

c. line 288: “are” -> “were”

This change was made.

d. line 436: Lh/L might be a reasonable approximation, but, as explained in my original review, it is not strictly true because the days do not have equal probabilities of inclusion in the sample (the days at the beginning and end of the study period are less likely to be included, unless you allow periods that "wrap" around to cover the beginning and the end, which is not very practical operationally). I think you need to explain this (and note that Borger et al. (2022) uses Lh/L too).

Thank you. You are correct that it is not appropriate to characterize Lh/L as the probability of selecting the particular set of days within the study recruitment period. We have revised the wording to instead describe this factor as an estimate of the proportion of new enrollees during the study recruitment period who enrolled during the set of recruitment days at the site. 

e. line 526-530: I have a different explanation/ suspicion about the estimates of flow rates, but your explanation is also possible. If it was just variability, I'd still expect a fit to be flat. But I think no further adjustment to the text is needed.

f. You response to the referees about weights says that you added a reference to a report with details. I looked at the report and it is minimally sufficient; I note that the weighting is described only in words and no equations are given. You should still consider providing additional details about the weighting process in the paper, especially how it was adapted to changes made in the field, since that is the key topic in your paper. For example, certainty units are not mentioned.

In this revision of the manuscript, we have provided additional details on the computation of the weights.

g. In Fig. 1 I don’t find it clear whether “160 units / 42 SAs” (and other similar) means that there are 202 things all together; I suspect not, but you could help with something like “160 units incl 42 SAs”.

The counts of SAs provided in the figure correspond to the number of SAs associated with the units. You are correct that these are not additional units. We have revised the figure to replace “## SAs” with “(in ## SAs)”.

---

## [Editor Report · Decision Letter 2]

4 May 2023

Implementation of a sample design for a survey of program participants using time-location sampling

PONE-D-21-33776R2

Dear Dr. Siegfried,

We’re pleased to inform you that your manuscript has been judged scientifically suitable for publication and will be formally accepted for publication once it meets all outstanding technical requirements.

Kind regards,

Paul A. Smith

Guest Editor

PLOS ONE

Additional Editor Comments (optional):

Thank you for the responses to the latest round of comments. I could wish that you had taken the opportunity to write a few sentences explicitly setting out the target population and the study population and the differences between them, earlier in the paper, but the text is now sufficiently clear about these differences. Likewise, Table 2 is very helpful in showing the ratio of achieved to expected sample of the target groups, but the text does not mention (even in passing) the effect on the design objectives, which I expect to be that target subgroups were relatively under-represented in the sample so that results for these subgroups are less accurate than anticipated. Nevertheless, there is now sufficient information for a reader to make an informed assessment about the consequences of the changes to the implementation of the design.

I particularly appreciated the summary of the weighting strategy. This version of the paper is now satisfactory.

Best wishes,

paul
---

## [Editor Report · Acceptance letter]

8 May 2023

PONE-D-21-33776R2 

Implementation of a sample design for a survey of program participants using time-location sampling 

Dear Dr. Siegfried:

I'm pleased to inform you that your manuscript has been deemed suitable for publication in PLOS ONE. Congratulations! Your manuscript is now with our production department. 

Kind regards, 

on behalf of

Professor Paul A. Smith 

Guest Editor

PLOS ONE